# Detailed systematic analysis of recruitment strategies in randomised controlled trials in patients with an unscheduled admission to hospital

Ceri Rowlands,[1] Leila Rooshenas,[1,2] Katherine Fairhurst,[1,2] Jonathan Rees,[2,3] Carrol Gamble,[4] Jane M Blazeby[1,2,3]

[1]MRC ConDuCT-II Hub for Trials Methodology Research, School of Social and Community Medicine, University of Bristol, Bristol, UK
[2]School of Population Health Sciences, Bristol Medical School, University of Bristol, Bristol
[3]Division of Surgery, Head and Neck, University Hospitals Bristol NHS Foundation Trust, Bristol, UK
[4]MRC North West Hub for Trials Methodology Research, Institute of Translational Medicine, University of Liverpool, Liverpool, UK

**Correspondence to**
Dr Leila Rooshenas;
Leila.rooshenas@bristol.ac.uk

## ABSTRACT

**Objectives** To examine the design and findings of recruitment studies in randomised controlled trials (RCTs) involving patients with an unscheduled hospital admission (UHA), to consider how to optimise recruitment in future RCTs of this nature.

**Design** Studies within the ORRCA database (Online Resource for Recruitment Research in Clinical TriAls; www.orrca.org.uk) that reported on recruitment to RCTs involving UHAs in patients >18 years were included. Extracted data included trial clinical details, and the rationale and main findings of the recruitment study.

**Results** Of 3114 articles populating ORRCA, 39 recruitment studies were eligible, focusing on 68 real and 13 hypothetical host RCTs. Four studies were prospectively planned investigations of recruitment interventions, one of which was a nested RCT. Most recruitment papers were reports of recruitment experiences from one or more 'real' RCTs (n=24) or studies using hypothetical RCTs (n=11). Rationales for conducting recruitment studies included limited time for informed consent (IC) and patients being too unwell to provide IC. Methods to optimise recruitment included providing patients with trial information in the prehospital setting, technology to allow recruiters to cover multiple sites, screening logs to uncover recruitment barriers, and verbal rather than written information and consent.

**Conclusion** There is a paucity of high-quality research into recruitment in RCTs involving UHAs with only one nested randomised study evaluating a recruitment intervention. Among the remaining studies, methods to optimise recruitment focused on how to improve information provision in the prehospital setting and use of screening logs. Future research in this setting should focus on the prospective evaluation of the well-developed interventions to optimise recruitment.

## Strengths and limitations of this study

► This review is the first to focus on the complex issue of recruitment to RCTs involving patients undergoing an unscheduled hospital admission (UHA).

► This review is the first publication to use the ORRCA database (Online Resource for Recruitment Research in Clinical TriAls) in generating recruitment research.

► The ORRCA database continues to evolve as updates encompass newly published recruitment research. Updates to the database may have generated further UHA research articles since the completion of this review.

the potential participant and the demanding working environment can complicate the process of identifying eligible patients, approaching them, and obtaining informed consent for trial participation. Patients are often in pain, unwell, and anxious about the underlying problem. There may also be time limitations due to the urgent need to deliver the clinical treatments under evaluation.[2 3]

There are a number of existing systematic reviews of methods to optimise recruitment to trials in a variety of clinical contexts and patient populations, including cancer,[4–6] primary care,[7] geriatrics[8 9] and minority community patients,[10] or a mix of clinical settings.[11–17] None specifically focus on recruitment of patients undergoing UHA.

Established methods exist for recruiting potential RCT participants who are unwell or unconscious and lack capacity. In these circumstances, permission for enrolment into an RCT may be sought from a surrogate decision-maker (SDM),[18] or through deferred consent (also known as 'exception from informed consent'), a process where a participant is recruited into the trial in order for urgent treatment to be provided

## INTRODUCTION

Recruitment to randomised controlled trials (RCTs) is the biggest obstacle to successful trial conduct.[1] Recruitment may be particularly challenging among patients who have an unscheduled hospital admission (UHA). In this situation, the clinical condition of

BMJ

and subsequently asked to provide written consent for ongoing participation once they regain capacity.[19] However, recruitment may be particularly challenging where patients are acutely unwell, but retain capacity to decide on enrolment into an RCT. Reviewing the literature on how to optimise recruitment in this setting may lead to valuable insights and identify areas where further research is needed.

The aim of this paper was to examine the design and findings of recruitment studies in RCTs involving patients with a UHA, to consider how to optimise recruitment in future RCTs of this nature.

## METHODS
### Search strategy
Articles were identified through manually screening each entry within the ORRCA recruitment research database (Online Resource for Recruitment research in Clinical triAls; http://www.orrca.org.uk/). The ORRCA project was funded by the UK Medical Research Council Hubs for Trials Methodology Research Network. It provides a comprehensive online database of published empirical and non-empirical papers about recruitment to clinical research. ORRCA is populated from an extensive systematic search of the Cochrane Library, MEDLINE (Ovid), SCOPUS, ERIC and SCI-EXPANDED and SSCI (via ISI Web of Science). The search strategy employed by ORRCA was based on a Cochrane systematic review of trial recruitment.[16] Further details about the formation of the ORRCA database is reported on their website (http://www.orrca.org.uk/). In this review, a full up-to-date copy of the ORRCA database was obtained in January 2016, and the database was searched in February 2016. At this stage, ORRCA contained publications relevant to recruitment published up to and including the end of December 2014. Publications from 2015 onwards were not available due to ongoing work in processing articles in the ORRCA database.

### Study eligibility criteria
Any study held within ORRCA that reported on recruitment to RCTs involving UHAs was eligible for inclusion. UHA was defined as an unscheduled admission to hospital at short notice because of clinical need. This included prehospital care, intensive care unit admissions, and accident and emergency (A&E) department attendances. Studies that reported on a mix of patients undergoing scheduled and unscheduled admissions were eligible if the findings for the UHA study population were described separately. Papers that reported recruitment strategies for a mixture of RCTs and other types of research (ie, non-RCTs) were only eligible for inclusion if the recruitment strategies for RCTs were described separately. RCTs that involved children (age <18 years) or patients with acute psychiatric illness were excluded, as these patients would not normally be subject to the usual recruitment processes due to differences in the consent processes. Systematic reviews of methods to optimise recruitment were scrutinised for relevant primary articles, but not included in the analysis. Editorials and studies of recruitment to non-RCTs were excluded. Abstracts were also excluded because these rarely included the necessary contextual information and data needed to make a meaningful contribution to the dataset for this study.

### Screening and selection process
One author (CR) screened all articles included within the ORRCA database. Duplicate screening was carried out by one other author (KF) on 10% of the database. KF was blinded to the original screening decisions. Papers were assessed at title and abstract level according to the eligibility criteria. Differences in opinion were resolved by discussion between CR and KF, and any remaining differences in opinion were referred to another member of the study team (JMB) if required. The aim was to reach agreement for screening decisions on all studies within this sample. Agreement was reached on 271/300 articles screened. Of the 29 discrepancies raised, 22 were resolved following discussion between CR and KF. The seven remaining papers were discussed with a third author (JMB), which resulted in two of these papers being included and five being excluded. No paper which was suggested to be included by the second reviewer was eventually included in the review.

We calculated a kappa statistic for the double screened articles above. Ten per cent of articles were double screened with a Kappa=0.677 (SE 0.048, P<0.001) suggesting 'good' agreement. As such, the remaining 90% of articles in the ORRCA database were screened by a single screener (CR). The third arbiter (JMB) involved in screening the 10% sample was also consulted for screening the remaining 90% of articles in the ORRCA database, in instances where the single screener (CR) was uncertain about an article's eligibility.

### Definitions: host RCT
All studies focused on recruitment to one or more specific 'host' RCTs. In this paper, a host RCT refers to the underlying randomised controlled trial (ie, addressing a clinical question) in which the recruitment of participants took place. A host RCT could be a pilot or a main trial. Some recruitment papers reported on community consultations in which the views of the public were sought to establish the likelihood of recruitment success or acceptability of a proposed trial. This approach is typically used when the study population may be critically ill at the time of recruitment (and therefore may be unable to provide full, written informed consent).[20] In recognition of this, a clear differentiation was made between studies that focused on recruitment to an existing clinical RCT (a 'real' host RCT) versus potential recruitment to an RCT that did not exist (a 'hypothetical' host RCT), but is proposed to exist in order to estimate its acceptability to potential participants. A 'recruitment study' refers to

**Table 1** Number of included studies, host RCTs and hypothetical RCTs categorised according to the new recruitment study types (A to D)

| Category | Recruitment study design | Recruitment studies (n=39) | Real host RCTs (n=68) | Hypothetical host RCTs (n=13) |
|---|---|---|---|---|
| A | RCTs of interventions to optimise recruitment nested within one or more host RCTs | 1 | 1 | NA |
| B | Prospectively designed, non-randomised studies of interventions to optimise recruitment to one or more host RCTs | 3 | 3 | NA |
| C | Studies describing recruitment experiences involving one or more host RCTs | 24 | 64* | NA |
| D | Studies to consider recruitment within proposed hypothetical RCTs (commonly known as community consultations) | 11 | NA | 13† |

*The 24 recruitment studies reported data from 64 real host RCTs, that is, a number of recruitment studies reported data from more than one real host RCT.
†The 11 recruitment studies reported data from 13 hypothetical studies, that is, two recruitment studies reported data from more than one hypothetical host RCT.
NA, not applicable—category does not apply to type of host RCT; RCT, randomised controlled trial.

research into the process of recruiting eligible participants, in the context of one or more real or hypothetical host RCTs.

## Definitions: recruitment study design
In order to group similarly designed recruitment studies together and enhance data analysis, a new categorisation system for different recruitment study designs was developed (Categories A to D). Consideration was given to the design of the recruitment study and whether a real or hypothetical host RCT was used. The categories are provided in table 1.

## Contacting study authors
If an appropriate recruitment study did not adequately describe the host trial, the study authors were contacted by email to determine whether the host trial met the eligibility criteria. Three attempts, each 2 weeks apart, were made to contact study authors. If no response was received, then the paper was excluded.

## Data extraction and synthesis
CR extracted the data using a predesigned and piloted data extraction form. Data extracted from eligible studies included clinical details of the underlying host RCT, the rationale for conducting a recruitment study, a summary of the recruitment study findings, recommendations for improving recruitment and suggestions for further research. Where specific recruitment interventions had been evaluated, further details regarding the interventions were collected, including the timing of information exchange, informed consent and randomisation. No statistical analyses were planned, as the review was expected to provide a descriptive analysis of results due to the anticipated heterogeneous nature of recruitment strategies presented. During the process, multiple meetings were undertaken with JMB and LR to examine papers and check data extraction processes as required.

## RESULTS
### Study selection
A total of 3114 articles were identified within the ORRCA database. After initial screening at title and abstract level, 3044 articles were excluded, leaving 70 potentially eligible for which full texts were obtained. A further 31 articles were excluded following full-text screening. Duplicate screening did not produce any discrepancies that could not be resolved through discussion. In total, 39 recruitment studies[21–58] were identified, which reported results from 68 real host RCTs and 13 hypothetical host RCTs and were included in this review.

A number of recruitment studies described results obtained from more than one real or hypothetical RCT (figure 1).

The 68 real host trials (around which recruitment was focused) were predominantly multicentre RCTs (63/68) with large study populations (median, 624 participants; range, 4–58 050) and typically evaluated non-invasive medical interventions (61/68) (table 2). The apparent predominance of RCTs in neurology is caused by one recruitment study that included data from 32 separate RCTs. With exception to this, the clinical settings of the host RCTs varied, encompassing several medical disciplines.

The majority of recruitment studies were simple descriptive non-randomised studies reporting previous experiences/challenges of recruitment to a host RCT (Category C, n=24). There were 11 that proposed a hypothetical RCT to a study population (Category D, n=11). Only one article used what is considered to be the optimal method for evaluating an intervention—a randomised comparison of a recruitment intervention nested within a host RCT (Category A). Three studies prospectively evaluated recruitment interventions using non-randomised study designs (Category B, table 1).

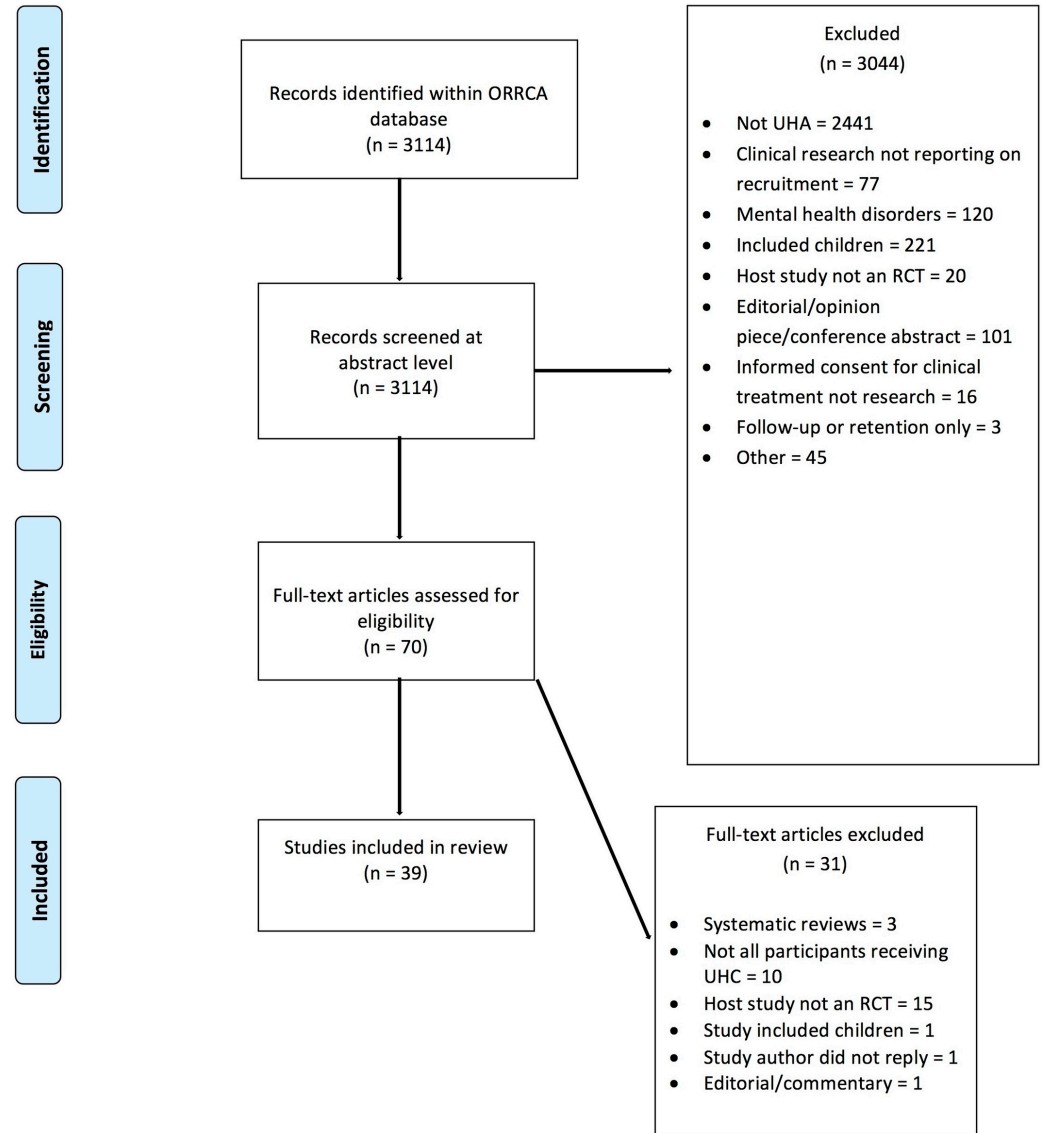

**Figure 1** Study selection PRISMA flow diagram. ORRCA, Online Resource for Recruitment Research in Clinical TriAls; PRISMA, Preferred Reporting Items for Systematic Review and Meta-Analysis; RCT, randomised controlled trial; UHA, unscheduled hospital admission; UHC, unplanned hospital care.

## RCTs of interventions to optimise recruitment nested within one or more host RCTs (Category A)

Only one of the included recruitment studies investigated two recruitment strategies using a randomised design (Category A studies, table 3). The rationale for this study was the limited time available for recruitment due to the acute medical treatment required by patients.[21] Patients randomised to the intervention group received 'advanced notification' of the trial (via fax or phone) designed to offer patients more time to consider trial participation, compared with the control group who only received information once they met with the clinical team. Consent to participate in the host RCT was obtained in 27/50 (54%) and 25/50 (50%) patients in the intervention and control groups, respectively (P=0.69). Although no improvement in overall recruitment rates was demonstrated using advance notification, the provision of early information was demonstrated to be feasible.

## Prospectively designed, non-randomised studies of interventions to optimise recruitment to one or more host RCTs (Category B)

The common rationale for this type of recruitment study design (Category B studies, table 3) was the limitations of time when dealing with patients who required acute treatment. The need for urgent treatment was thought to hinder the ability of the study team to gain informed consent (IC) for trial participation.

The strategy of optimising information provision in the prehospital setting used in the one Category A study was mirrored by two Category B studies, which used the presence of prehospital staff to engage potential trial participants.[22] [24] Recruitment in these studies was reportedly

**Table 2** Summary characteristics of the host RCTs and hypothetical host RCTs in this review

| Trial characteristic | Real RCTs (n=68) | Hypothetical RCTs (n=13) | Total (n=81) |
|---|---|---|---|
| Clinical setting | | | |
| Neurology | 39* | 4 | 43 |
| Cardiology | 8 | 4 | 12 |
| Obstetrics | 3 | 1 | 4 |
| Infection | 3 | 0 | 3 |
| Trauma | 6 | 3 | 9 |
| Critical care | 9 | 1 | 10 |
| Type of interventions | | | |
| Invasive/non-invasive | 5/61 | 4/9 | 9/70 |
| Unknown | 2 | 0 | 2 |
| Trial design | | | |
| Main RCT/pilot RCT | 65/3 | n/a | 65/3 |
| Two groups/more than two groups | 63/5 | 11/2 | 74/7 |
| No of centres† | | | |
| Single centre/multicentre/unknown | 3/63/2 | 0/4/9 | 3/67/11 |
| <20 centres | 18 | | |
| 20–100 centres | 33 | | |
| >100 centres | 15 | | |
| Median no of centres (range) | 45 (1–818) | | |
| No of participants† | | | |
| <500 | 22 | | |
| 500–1500 | 32 | | |
| >1500 | 14 | | |
| Median no of participants (range) | 624 (4–58 050) | | |

*One recruitment paper included data from 32 stroke host RCTs.

†Hypothetical RCT data did not include information beyond single or multicentre, nor the suggested number of participants.

RCT, randomised controlled trial.

optimised through provision of brief verbal information to participants as they travelled to hospital, and initial verbal consent that sought permission to deliver the emergency trial intervention. Further information was provided and written consent was subsequently obtained when the patients were stabilised, in hospital. One of these studies[24] also provided training to prehospital staff to improve their understanding of trial conduct, and devised a simple assessment of capacity to ensure that patients' initial verbal consent was valid.

The remaining prospectively designed (Category B) study focused on using technology to enhance recruitment during an influenza outbreak.[23] As patient numbers would be expected to rise rapidly across a wide geographical area, the study team devised a system which provided them with automated, real-time alerts whenever an eligible participant was identified in each hospital. This allowed one centralised study team to cover numerous study sites, enhancing recruitment opportunities. Although none of these studies provided numerical evidence of the effectiveness of their recruitment strategies, all authors concluded that their presented strategies were feasible and acceptable for use in UHA RCTs.

### Studies describing recruitment experiences involving one or more host RCTs (Category C)

Rationales for reporting authors' experiences of recruitment were similar to those in prospectively designed studies, including the limited time available for consent (n=13), and recruitment difficulties caused by the clinical condition of the patients (n=18) (Category C studies, table 4). Some studies were prompted by a host trial encountering recruitment difficulties (n=3). Of the 24 non-randomised studies describing recruitment experiences involving one or more host RCTs, 14 were observational and five were qualitative studies.

These studies reported experiences of trial participants and SDMs, or extracted verbatim information from written patient information sheets. Recommendations for optimising recruitment also mirrored Category A and B studies, highlighting the acceptability of verbal information provision and consent, or deferring consent altogether until an unwell patient is suitably stabilised. Additional benefits were seen in RCTs that used data from screening logs to uncover recruitment barriers and trials that performed regular site visits.[29–31 33 35 38 39 44] A subsection of qualitative studies, although not presenting recommendations for future trial conduct, highlighted their findings that many patients or SDMs who had provided consent to participate in an RCT did not recall much of the information provided to them during the consent process, suggesting that work was needed to improve consent in this setting.[31 32 40]

### Studies to consider recruitment within proposed hypothetical RCTs (commonly known as community consultations) (Category D)

Eleven studies reported community views about proposed 'hypothetical' RCTs (Category D studies, table 5). Although the rationales for conducting the studies were similar to studies involving real host RCTs (unwell patients and lack of time for consent), the study designs were varied including questionnaire surveys, interviews and focus group meetings. Verbal information provision, verbal consent and recruitment in the prehospital setting were identified as helpful recruitment strategies in these studies. However, these articles raised new issues around the appropriateness of using of SDMs when patients are too unwell to provide consent for themselves and raised additional issues around who the SDM should be (next of kin (NOK) or an available physician). Some studies found a preference for the use of SDMs,[50 52 56] while others expressed that NOK or physicians should not be used for their specific hypothetical RCTs.[53 55 58]

**Table 3** Rationale and types of recruitment interventions reported in Category A or B studies (ie, prospectively nested within RCTs)

| Author and year | Rationale(s) | Descriptive summary of recruitment intervention(s) | Classification of recruitment intervention | Timing of information provision regarding host RCT | Timing of informed consent for host RCT | Timing of randomisation for host RCT |
|---|---|---|---|---|---|---|
| Prospectively designed RCTs of interventions to optimise recruitment nested within one or more host RCTs (Category A studies) | | | | | | |
| Leira et al[21] 2009* | Limited time for IC in host RCT due to the clinical condition requiring urgent treatment | Intervention group: 'advance notification' about host RCT using phone and fax prior to meeting prehospital team Control group: no information prior to meeting prehospital team | Advance notification using phone and fax | Intervention group: 'advance notification' prior to face-to-face meeting with prehospital team Control group: during face-to-face meeting with prehospital team | Intervention group: during face-to-face meeting with prehospital team Control group: during face-to-face meeting with prehospital team | Intervention group: while patient in prehospital setting Control group: while patient in prehospital setting |
| Prospectively designed, non-randomised studies of interventions to optimise recruitment to one or more host RCTs (Category B studies) | | | | | | |
| Chow et al[23] 2011 | Recruitment team would be overwhelmed by covering multiple sites over a wide geographical area | An automated service notified recruitment team in real time when a potentially eligible participant was identified across multiple centres | Automated service to facilitate real-time notifications to recruitment team covering multiple sites | During face-to-face meeting with recruitment staff (once eligibility confirmed) | During face-to-face meeting with recruitment team | While patient in hospital |
| Shaw et al[24] 2014 | Limited time for IC in host RCT due to the clinical condition requiring urgent treatment Prehospital staff have limited experience of RCTs | Standardised verbal information was provided by prehospital staff before a simple capacity assessment, verbal consent and delivery of emergency treatment Host RCT procedures were tailored towards prehospital staff, who also attended a 1-day training event | Optimising information provision in prehospital setting Allowing verbal consent from patient or SDM Training of prehospital staff | Optimising information provision in prehospital setting Further full information provided in hospital | Initial verbal consent given in prehospital setting Written informed consent gained in hospital | While patient in prehospital setting |
| Beshansky et al[22] 2014 | Limited time for IC in host RCT due to the clinical condition requiring urgent treatment | Standardised verbal information was provided by prehospital staff before a simple capacity assessment, verbal consent and delivery of emergency treatment | Optimising information provision in prehospital setting Allowing verbal consent from patient | Optimising information provision in prehospital setting Further full information provided in hospital | Initial verbal consent in prehospital setting Written informed consent in hospital | While patient in prehospital setting |

*Leira et al—patients were randomised to the *recruitment* intervention or control group prior to the recruitment team seeing the patient, and without consent from the participant.
IC, informed consent; RCT, randomised controlled trial; SDM, surrogate decision-maker.

**Table 4** Frequency of rationales, study designs and recommendations from non-randomised studies describing recruitment experiences involving one or more host RCTs (ie, Category C studies)

| Recruitment study characteristic | Description | Frequency in Category C recruitment studies (n=24)* |
|---|---|---|
| Rationale | Patients too unwell to provide IC | 18 |
| | Limited time for IC in host RCT due to the clinical condition requiring urgent treatment | 13 |
| | Host RCT not meeting recruitment targets (at one or more sites) or terminated due to poor recruitment | 3 |
| | To better understand the impact of altering eligibility criteria on recruitment | 2 |
| | To better understand the impact of availability of SDMs on recruitment | 2 |
| | To better understand the recruitment process in a host RCT | 1 |
| | To better understand clinicians reasons for refusing patient participation in host RCT | 1 |
| Recruitment study design | Observational study of recruitment | 14 |
| | Qualitative studies of host participants/SDMs or PIS | 5 |
| | Survey of host RCT participants | 2 |
| | Survey of clinical staff involved in host RCT | 1 |
| | Simulation study evaluating the effect of altering eligibility criteria in multiple host RCTs | 1 |
| | Meta-analysis of recruitment data in host RCTs | 1 |
| Recommendations for optimising recruitment in future RCTs or areas for further research into recruitment† | To provide RCT information verbally and allow a verbal consent process | 10 |
| | To use a screening log can to provide insight into recruitment difficulties | 8 |
| | Patients or SDM were unable to recall key RCT information after providing IC† | 5 |
| | To use a 'waiver of consent'/'deferred consent'/'EFIC' | 4 |
| | To perform regular site visits | 3 |
| | To use a broad eligibility criteria/broad therapeutic window | 2 |
| | To use SDMs | 2 |
| | Novel methods for obtaining IC are required† | 2 |
| | To replace poorly recruiting centres | 1 |
| | To approach more eligible patients | 1 |
| | To survey staff involved with host RCT to provide insight into recruitment difficulties | 1 |

*Each study may appear more than once in the relevant characteristics section (eg, if it described >1 rationale or produced >1 finding/recommendation).
†Items for further research and not recommendations for optimising recruitment.
EFIC, exception from informed consent; IC, informed consent; PIS, patient information sheet; RCT, randomised controlled trial; SDM, surrogate decision-maker.

## DISCUSSION

This review aimed to examine and summarise studies and methods used to optimise recruitment in RCTs in patients with an UHA. It had the purpose of using the information to consider how to optimise recruitment in this challenging clinical setting in future studies. In the ORCCA database of recruitment research, only 39 out of a possible 3114 articles (1.25%) focused on recruitment to RCTs in the UHA setting. Only one of these studies was a randomised comparison of recruitment strategies; the majority of studies consisted of simple study designs describing recruitment experiences. Eleven further studies involved hypothetical RCTs, and while of some value, it is uncertain how these types of investigations translate into optimal RCT design. This work therefore highlights the need for development of interventions to optimise recruitment in the UHA setting and prospective evaluation of their effectiveness and acceptability.

**Table 5** Frequency of rationales, study designs, main findings and recommendations from non-randomised studies designed study to consider recruitment within proposed hypothetical RCTs (ie, Category D studies, commonly known as community consultations)

| Recruitment study characteristic | Description | Frequency in Category D recruitment studies (n=11)* |
|---|---|---|
| Rationale | Patients too unwell to provide IC | 9 |
| | Limited time for IC in host RCT due to the clinical condition requiring urgent treatment | 6 |
| | To explore the accuracy of decisions made by NOK when acting as SDM | 1 |
| Recruitment study design | Questionnaire survey | 5 |
| | Face-to-face interview | 4 |
| | Telephone survey | 1 |
| | Focus group meetings | 1 |
| Recommendations for optimising recruitment in future RCTs or main findings† | To use a physician as a SDM | 4 |
| | To use NOK as a SDM | 4 |
| | To use EFIC | 3 |
| | Not to use NOK as a SDM | 2 |
| | To provide RCT information verbally and allow a verbal consent process | 1 |
| | To allow recruitment in prehospital setting | 1 |
| | To perform community consultations to estimate host RCT recruitment rates† | 1 |
| | Not to use a physician as a SDM | 1 |
| | Not to use EFIC | 1 |
| | To perform community consultations to aid selection of relevant study outcomes† | 1 |

*Each study may appear more than once in the relevant characteristics section (eg, if it described >1 rationale or produced >1 finding/recommendation).
†Items reported as main findings, but not recommendations for optimising recruitment.
EFIC, exception from informed consent; IC, informed consent; NOK, next of kin; RCT, randomised controlled trial; SDM, surrogate decision-maker.

### Comparison to existing literature

There have been several reviews that have summarised the evidence for optimising recruitment in trials in other clinical conditions or contexts, including cancer,[4–6] geriatrics,[8 9] primary care[7] and a mix of clinical settings.[11–17] Similar to our findings, these reviews have commonly highlighted the lack of high-quality evaluations of recruitment interventions.[6 16] Despite this, some of the reviews have identified effective recruitment strategies, although these vary in the extent to which they are likely to be transferrable to the UHA setting. For example, it is unclear if interventions such as telephone reminders,[16 17] 'education sessions' about the health condition[11] and use of monetary incentives[11 17] are as effective or appropriate in the UHA context, given the specific factors that may compromise recruitment in this setting (eg, patients in pain/distress, short timeframes for recruitment, busy settings etc). Other reviews have drawn attention to interventions aimed at recruiters, such as appropriate training/guidance,[12] reduction of clinical workload[7 12] and 'research protected' time.[12] These have potential to be helpful in trials conducted in the UHA setting, although further research is needed to examine the content/nature of the training materials needed and the logistics of implementing these types of interventions. Finally, some reviews have shown that features of RCT study design—such as open rather than placebo-controlled trials—are associated with better recruitment outcomes.[16 17] Recommendations that advise against particular study designs may limit the quality of evidence generated to guide future patient care and dissuade from the most important (and appropriate) clinical questions from being addressed. Furthermore, there is a growing body of evidence to indicate that it is possible to recruit to more complex RCTs with appropriate training and support.[59] More generally, use of integrated qualitative research to understand and address recruitment difficulties is being increasingly recommended in more recent reviews[12 60] and is likely to inform novel insights if applied to trials in UHA settings.

Some reviews and individual studies have produced recommendations that are likely to be particularly relevant for developing UHA-specific recruitment strategies for future evaluation. These strategies may tackle some of the context-specific difficulties that are likely to be experienced in UHA settings. For example, one systematic review focusing on recruitment to RCTs involving patients with cancer or organ failure highlighted the potential for providing audiovisual information (such as a video to explain the RCT) to facilitate RCT recruitment.[59] Such an approach may be helpful in the UHA setting, given that large amounts of written information may not be appropriate in patients who are in pain or feeling distressed. This recommendation is also in keeping with guidance issued by the National Health Service Health Research Authority, which suggests using alternatives to written information.[61]

A verbal exchange of information and providing initial verbal consent has been previously suggested as a preferred strategy to written alternatives in RCTs taking place in the emergency setting.[62] It is agreed that for UHAs, this could be preferable, although it is considered that further work is needed to develop this type of verbal consent to ensure that quality assurance is still achieved for consent. Another suggested solution to this problem is the use of an independent patient advocate, who may oversee such conversations between trial team members and acutely unwell patients, to verify that appropriate information exchange took place and to act as an assessor of a patient's willingness to participate in the RCT.[63] This strategy provides one potential solution to a significant obstacle in recruitment of patients undergoing an UHA. However, this may be practically difficult to achieve given that UHAs can occur at any time of the day or night and such trained patient advocates would also need to be available during these times, making a trial more expensive.

Finally, a non-systematic review article focusing on recruitment to emergency medicine research also highlighted a similar problem in a lack of high-quality evidence on recruitment. It suggested support for the use of deferred consent (also known as exception from informed consent), which was highlighted in this review and additionally raised the issue that the use of SDMs for consent may be problematic due to the pressures of time and the emotional stress family members will be under while a relative is acutely unwell.[3]

### Strengths and weaknesses of this study

This review is the first to systematically focus on recruitment strategies in the UHA setting. With a reported growing number of UHAs presenting great challenges to modern-day healthcare provision, the conduct of RCTs in the UHA will inevitably develop as an area of research.

The review may be limited through its reliance on a single search of the ORRCA database, conducted in February 2016, at which point the database contained publications relevant to recruitment published up to the end of December 2014. The ORRCA database continues to evolve as updates encompass newly published recruitment research. Updates to the database may have generated further UHA research articles since the search for this review. It is possible that any new update could make an important and significant contribution to this field because so little has been done in this area thus far.

The review may be limited because a single researcher reviewed the majority of the ORRCA entries, and it is possible that ORRCA may not have included all relevant articles in the first place. This work may also be at risk of publication bias, as we chose to exclude abstracts based on the assumption that these were unlikely to include all the data items we were interested in. Excluding abstracts may have resulted in omission of some potentially valuable information. Although some authors recommend the inclusion of conference abstracts within a review,[64 65] there is some evidence to suggest that there is discordance between the content of abstracts and the subsequent full-text publication, and as such, including abstracts may introduce unreliable data.[66 67] Another weakness is that the majority of recruitment studies were retrospective analyses of processes and events that occurred during the host RCT. These data were not necessarily collected with the intention of evaluating RCT recruitment strategies. This may limit the quality of the data and the use of the recommendations arising from the included studies.

Finally, a risk of bias assessment of the included recruitment studies was not performed because only one of the recruitment studies was an RCT (ie, a randomised, controlled evaluation of a recruitment intervention).

### Unanswered questions and future research

Some articles within this review demonstrated inconsistent conclusions about the value of SDMs, who these should be and how these should operate. Future research should examine these issues in more depth, in a variety of clinical contexts, focusing on the roles of SDMs in different RCTs.

It is uncertain how well the results of studies using hypothetical RCTs can be translated to the conduct of real RCTs. There may be some validity in the findings from hypothetical RCTs, as highlighted in this review by the similarity of the results from studies that used real RCTs and hypothetical RCTs. Further work is required to ascertain the extent to which findings from studies that use hypothetical RCTs influence the design and conduct of real RCTs.

As part of this review, a classification system was devised in order to group together similarly designed recruitment research studies. Before any such classification system could be used more widely, it would require validation by testing its applicability to at least one further set of

recruitment papers, preferably in a different healthcare setting.

This review highlighted strategies to deal with patients in the prehospital setting who required urgent treatment. However, not all patients who are cared for in the prehospital setting require treatment immediately. Some may require transport to hospital for further assessment and potential treatment. Further research could explore whether providing early trial information to such patients, based on their symptoms or presumed diagnosis, could affect trial recruitment should treatment be required later. This strategy could prove to be useful in a broader range of unscheduled hospital admissions.

Although some recommendations for optimising recruitment could be drawn from this review, the overall lack of research in this area, particularly among high-quality, methodologically robust studies, is a limiting factor. Future recruitment studies in this clinical setting should focus on studies with higher methodological rigour, by developing novel interventions to optimise recruitment and prospectively evaluating their effectiveness through an appropriate study design.

## CONCLUSION

There is a relative paucity of high-quality research on strategies to optimise recruitment to RCTs involving unscheduled hospital admissions. Some emerging recommendations include optimising information provision about the trial in the prehospital setting to improve recruitment where treatment is required urgently, or using technology to facilitate recruitment across many hospital sites. Screening log data can also provide useful insight to specific barriers to recruitment. Future research in this setting should focus on conducting studies with higher methodological rigour, by developing interventions to optimise recruitment and prospectively evaluating their effectiveness.

**Acknowledgements** The following are gratefully acknowledged for their assistance and support with this work: Shaun Treweek, Anna Kearney, Nicola Harman.

**Contributors** CR, LR, JR and JMB designed the study and formulated the protocol. CR and KF performed all data extraction, which was reviewed and critically analysed by CR, LR and JMB. CG facilitated access to the ORRCA database and critically appraised the review. CR produced the first draft of the manuscript and LR and JMB made substantial contributions to the formulation of the final version. All authors read and approved the final manuscript.

**Funding** CR is supported by the Medical Research Council (MRC) Hub for Trials Methodology Research Network and the Collaboration and Innovation for Difficult Trials in Invasive Procedures (ConDuCT-II) Hub for Trials Methodology Research (MR/K025643/1). The funders had no role in the study design, data collection and analysis, decision to publish or preparation of the manuscript.

**Competing interests** None declared.

**Patient consent** Obtained.

**Provenance and peer review** Not commissioned; externally peer reviewed.

**Data sharing statement** As this is a review of published literature, there are no additional data available.

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
