## [Reviewer comments · BMJ Open]

ARTICLE DETAILS

TITLE (PROVISIONAL)	A detailed systematic analysis of recruitment strategies in randomised controlled trials in patients with an unscheduled admission to hospital
AUTHORS	Rowlands, Ceri; Rooshenas, Leila; Fairhurst, Katherine; Rees, Jonathan; Gamble, Carrol; Blazeby, Jane

VERSION 1 – REVIEW

REVIEWER	David Torgerson York Trials Unit, University of York, UK
REVIEW RETURNED	21-Jul-2017

GENERAL COMMENTS	A useful summary of the recruitment literature for trials of interventions among unplanned hospital admission patients. The paper was well written and easy to follow – aims and objectives were clear I only have a few minor comments below. In abstract first sentence define: UHA Screening and selection process (page 5): shouldn't a kappa statistic be calculated for the 10% double screened and then if it is high double screening of the remaining 90% isn't required conversely if low it is. You go on to mention in the results that duplicate screening didn't produce any discrepancies that couldn't be resolved through discussion – but the issue relates to the other 90% you didn't duplicate screen. It is rather irrelevant whether you can resolve the issue on the sample (unless it is a trivial thing that applies to other studies that the single screener can apply). This is a common mistake that people make when quality assuring reviews.
---

REVIEWER	Christopher Jones, MD Cooper Medical School of Rowan University, USA I am an investigator on unrelated studies sponsored by Roche Diagnostics, AstraZeneca, and Janssen.
REVIEW RETURNED	11-Aug-2017

GENERAL COMMENTS	Thank you for the opportunity to review your manuscript, "A detailed systematic analysis of recruitment strategies in randomised controlled trials in patients with an unscheduled admission to hospital." Your systematic review covers a very important topic, which is likely to be of interest across a broad range of clinical specialties.
--

While the lack of evidence within the primary literature unfortunately limits the direct guidance that you can offer to investigators based on the results of your review, the manuscript does highlight the need for additional investigation into this important topic. Please consider these suggestions for your manuscript.

My biggest concern as this analysis stands is that the included data are already quite dated. The authors report that the ORRCA database contained publications up to 2014 at the time it was queried. While with any systematic review there will of course be some elapsed time between the literature search and the publication date, in this case the data are already over 3.5 years old. In my opinion, for this manuscript to really be a useful resource for trialists, this interval should be tightened up. This could be accomplished by either re-querying ORRCA, assuming that it has since been updated, or by designing a search of Medline and one or two other relevant databases for just the years 2014-2017.

It would be preferable to have had duplicate, independent screening of each ORRCA entry, rather than simply screening 10% of the sample in duplicate. Was the second screener blinded to the initial screener's classification decisions? How many cases within the 10% sample were flagged for possible inclusion by the second screener but not the first and then went on to be included in your final cohort of included studies?

Two issues confused me a bit as I read your paper. First, you describe the objective of the study "to examine and summarise recruitment details of RCTs undertaken in patients with a UHA...." This makes it sound as if you intended to review all RCT manuscripts to describe the recruitment strategies listed within each of their methods sections. I think it would be more accurate to state that you intended to identify and examine studies which evaluated methods of recruiting patients into RCTs which involve patients with UHAs. Second, at times it was difficult to keep the distinction between "recruitment studies" and "Real RCTs" in mind. This was particularly true when reading your tables. I would favor deleting the "Real host RCT" and "Hypothetical host RCT" columns from Table 1, as this information is already contained within your study design categories, and I found the additional columns to be distracting.

I'm not sure why you chose to exclude abstracts, as identifying an exhaustive list of studies which are relevant to the topic at hand, even when they did not result in publication of a full manuscript, is an important means of limiting publication bias and ensuring that a systematic review is as thorough as possible. Instead, I would recommend excluding studies on the basis of whether or not the information you need is available rather than the format of the publication.

You note the existence of a number of previous reviews which have addressed recruitment challenges within different patient populations. Given the limited evidence you found allowing for direct recommendations within your target population, it would be helpful within your discussion to summarize some of the specific evidence-based recommendations or recruitment techniques from these other populations which might theoretically be applicable to patients with an unscheduled hospital admission.

	Within your Limitations section you state that given the small proportion of eligible papers in the ORRCA database, it is unlikely that a significant body of research has been omitted due to the relatively dated nature of the search. I don't think this is a reasonable conclusion to draw, as the relatively weak literature base which you discovered means that even a handful of recently published, high-quality papers on this topic could potentially dramatically alter your conclusions. Your Limitations section should also be expanded to address several additional issues. First, it is possible that you may have missed relevant publications, particularly given that a single investigator reviewed the substantial majority of ORRCA entries, and that ORRCA may not have included all relevant articles in the first place. Additionally, you did not perform a formal quality or risk of bias analysis for the included studies. While doing so would be fairly challenging given the atypical nature of the papers you studied, this should still be noted as a limitation. Finally, studies of the type you included are likely at fairly high risk of publication bias, as these tend to be retrospective studies or secondary analyses of trial data collected for other reasons. This potential source of bias should also be noted.
--	---

VERSION 1 – AUTHOR RESPONSE

Reviewer: 1

1. A useful summary of the recruitment literature for trials of interventions among unplanned hospital admission patients. The paper was well written and easy to follow – aims and objectives were clear I only have a few minor comments below.

Response

Thank you for your comments

2. In abstract first sentence define: UHA

Response

As described above we have amended the abstract by replacing 'UHA' with 'unplanned hospital admission' in the first sentence.

3. Screening and selection process (page 5): shouldn't a kappa statistic be calculated for the 10% double screened and then if it is high double screening of the remaining 90% isn't required conversely if low it is.

Response

Thank you for highlighting this. We have calculated the Kappa statistics for the 10% of double screened abstracts which was 0.677 (SE= 0.048, $p < 0.0001$) suggesting adequate agreement[1]. On this basis, it was felt that double screening of the remaining abstracts was not indicated.

We have edited the 'screening and selection' sub-section of the Methods section to report this value and include this sentence, as follows:

“Screening and selection process

One author (CR) screened all articles included within the ORRCA database. Duplicate screening was carried out by one other author (KF) on 10% of the database. KF was blinded to the original screening decisions. Papers were assessed at title and abstract level according to the eligibility criteria. Differences in opinion were resolved by discussion between CR and KF, and any remaining differences in opinion were referred to another member of the study team (JMB) if required. The aim was to reach agreement for screening decisions on all studies within this sample. Agreement was reached on 271/300 articles screened. Of the 29 discrepancies raised, 22 were resolved following discussion between CR and KF. The seven remaining papers were discussed with a third author (JMB), which resulted in two of these papers being included and five being excluded. No paper which was suggested to be included by the second reviewer was eventually included in the review.

We calculated a kappa statistic for the double screened articles above. Ten per cent of articles were double screened with a Kappa = 0.677 (SE= 0.048, $p < 0.001$) suggesting “good” agreement. As such, the remaining 90% of articles in the ORRCA database were screened by a single screener (CR). The third arbiter (JMB) involved in screening the 10% sample was also consulted for screening the remaining 90% of articles in the ORRCA database, in instances where the single screener (CR) was uncertain about an article’s eligibility.”

4. You go on to mention in the results that duplicate screening didn’t produce any discrepancies that couldn’t be resolved through discussion – but the issue relates to the other 90% you didn’t duplicate screen. It is rather irrelevant whether you can resolve the issue on the sample (unless it is a trivial thing that applies to other studies that the single screener can apply). This is a common mistake that people make when quality assuring reviews.

Response

Thank you for pointing this out. The screening results papers which were discrepant in the double screening component of this work were resolved by discussion and a 3rd arbiter was used to facilitate this process (JMB). This identified specific issues which informed the remainder of the screening undertaken by a single screener (CR). In addition, the same arbiter was employed for the single researcher screening when it was unclear whether a paper should be excluded or included.

As described above, we have amended the screening and selection process sub- section of the methods section in order to address the helpful points raised.

Reviewer: 2

1. My biggest concern as this analysis stands is that the included data are already quite dated. The authors report that the ORRCA database contained publications up to 2014 at the time it was queried. While with any systematic review there will of course be some elapsed time between the literature search and the publication date, in this case the data are already over 3.5 years old. In my opinion, for this manuscript to really be a useful resource for trialists, this interval should be tightened up. This could be accomplished by either re-querying ORRCA, assuming that it has since been updated, or by designing a search of Medline and one or two other relevant databases for just the years 2014-2017.

Response

Thank you for this comment. We have amended the wording used throughout the manuscript to clarify that our review included articles up until the end of December 2014. Please see the response to the editorial comments point 1.

2. It would be preferable to have had duplicate, independent screening of each ORRCA entry, rather than simply screening 10% of the sample in duplicate. Was the second screener blinded to the initial screener's classification decisions? How many cases within the 10% sample were flagged for possible inclusion by the second screener but not the first and then went on to be included in your final cohort of included studies?

Response

As suggested, we have now included additional detail to address these points in the revised manuscript. Please see response to reviewer 1 points 3 and 4. Further information in relation to the specific points raised in the comment above are addressed below:

- Was the second screener blinded to the initial screener's classification decisions?

The first and second screeners worked independently and were not aware of each other's decisions. This is now described in the revised manuscript (as specified in the response to reviewer 1 (point 4).

- How many cases within the 10% sample were flagged for possible inclusion by the second screener but not the first and then went on to be included in your final cohort of included studies?

None of the double screened articles were included where reviewer 1 suggested exclusion and reviewer 2 suggested inclusion. Paragraph 3 of the methods section has been changed to clarify this, as shown in the response to reviewer 1 point 4.

3. Two issues confused me a bit as I read your paper. First, you describe the objective of the study "to examine and summarise recruitment details of RCTs undertaken in patients with a UHA..." This makes it sound as if you intended to review all RCT manuscripts to describe the recruitment strategies listed within each of their methods sections. I think it would be more accurate to state that you intended to identify and examine studies which evaluated methods of recruiting patients into RCTs which involve patients with UHAs.

Response

Thank you for your helpful comments. The abstract has been amended, as shown below:

"Objectives: To examine the design and findings of recruitment studies in RCTs involving patients with an unscheduled admission to hospital (UHA), to consider how to optimise recruitment in future RCTs of this nature."

4. Second, at times it was difficult to keep the distinction between "recruitment studies" and "Real RCTs" in mind. This was particularly true when reading your tables. I would favor deleting the "Real host RCT" and "Hypothetical host RCT" columns from Table 1, as this information is already contained within your study design categories, and I found the additional columns to be distracting.

Response

Thank you for this suggestion. Considering the table, we included both the numbers of recruitment studies and the number of real host RCTs to highlight to the reader that individual recruitment studies often used data from more than one real host RCT. We have placed an additional explanation in the footnote of Table 1 to highlight this in the revised manuscript. The amended version of Table 1 is shown at the end of this document, with the new additions highlighted in yellow.

We have also re-iterated this issue by adding a sentence in the results section of the manuscript, to reinforce explanations of the data presented in the table. The new text is shown below:

"Study selection

In total, 39 recruitment studies [2-39] were identified which reported results from 68 real host RCTs and 13 hypothetical host RCTs and were included in this review (Figure 1). A number of recruitment studies described results obtained from more than one real or hypothetical RCT.”

5. I'm not sure why you chose to exclude abstracts, as identifying an exhaustive list of studies which are relevant to the topic at hand, even when they did not result in publication of a full manuscript, is an important means of limiting publication bias and ensuring that a systematic review is as thorough as possible. Instead, I would recommend excluding studies on the basis of whether or not the information you need is available rather than the format of the publication.

Response

We decided to exclude abstracts because they did not contain sufficient information about the recruitment interventions, outcomes and host RCTs to make meaningful contribution to the dataset. We have clarified this in paragraph 2 of the methods section with this additional sentence:

“Study eligibility criteria

Abstracts were also excluded because these rarely included the necessary contextual information and data needed to make a meaningful contribution to the dataset for this study.”

6. You note the existence of a number of previous reviews which have addressed recruitment challenges within different patient populations. Given the limited evidence you found allowing for direct recommendations within your target population, it would be helpful within your discussion to summarize some of the specific evidence-based recommendations or recruitment techniques from these other populations which might theoretically be applicable to patients with an unscheduled hospital admission.

Response

We have updated the discussion to include more information and summarise the existing evidence from reviews although they did not include UHA patients. The updated text is below.

“Comparison to existing literature

There have been several reviews that have summarised the evidence for optimising recruitment in trials in other clinical conditions or contexts, including cancer (4-6), geriatrics (8,9) primary care (7), and a mix of clinical settings.(11-17). Similar to our findings, these reviews have commonly highlighted the lack of high quality evaluations of recruitment interventions (6,16). Despite this, some of the reviews have identified effective recruitment strategies, although these vary in the extent to which they are likely to be transferrable to the UHA setting. For example, it is unclear if interventions such as telephone reminders (16,17), ‘education sessions’ about the health condition (11), and use of monetary incentives (11,17) are as effective or appropriate in the UHA context, given the specific factors that may compromise recruitment in this setting (e.g. patients in pain/distress, short timeframes for recruitment, busy settings, etc.). Other reviews have drawn attention to interventions aimed at recruiters, such as appropriate training/guidance (12), reduction of clinical workload (7,12), and ‘research protected’ time (12). These have potential to be helpful in trials conducted in the UHA setting, although further research is needed to examine the content/nature of the training materials needed, and the logistics of implementing these types of interventions. Finally, some reviews have shown that features of RCT study design – such as open, rather than placebo-controlled trials - are associated with better recruitment outcomes (16,17). Recommendations that advise against particular study designs may limit the quality of evidence generated to guide future patient care, and dissuade from the most important (and appropriate) clinical questions from being addressed.

Furthermore, there is a growing body of evidence to indicate that it is possible to recruit to more complex RCTs with appropriate training and support.⁽⁵⁹⁾ More generally, use of integrated qualitative research to understand and address recruitment difficulties is being increasingly recommended in more recent reviews ⁽¹²⁾⁽⁶⁰⁾, and is likely to inform novel insights if applied to trials in UHA settings.“

7. Within your Limitations section you state that given the small proportion of eligible papers in the ORRCA database, it is unlikely that a significant body of research has been omitted due to the relatively dated nature of the search. I don't think this is a reasonable conclusion to draw, as the relatively weak literature base which you discovered means that even a handful of recently published, high-quality papers on this topic could potentially dramatically alter your conclusions.

Response

Thank you for this feedback. We have amended this comment in paragraph 2 of the 'strengths and weaknesses of this study' section to indicate that a small number of more recently published papers might make important contributions to the literature. The amended text is shown below:

“The review may be limited through its reliance on a single search of the ORRCA database, conducted in February 2016, at which point the database contained publications relevant to recruitment published up to the end of December 2014. The ORRCA database continues to evolve as updates encompass newly published recruitment research. Updates to the database may have generated further UHA research articles since the search for this review. It is possible that any new update could make an important and significant contribution to this field because so little has been done in this area thus far.”

8. Your Limitations section should also be expanded to address several additional issues. First, it is possible that you may have missed relevant publications, particularly given that a single investigator reviewed the substantial majority of ORRCA entries, and that ORRCA may not have included all relevant articles in the first place. Additionally, you did not perform a formal quality or risk of bias analysis for the included studies. While doing so would be fairly challenging given the atypical nature of the papers you studied, this should still be noted as a limitation. Finally, studies of the type you included are likely at fairly high risk of publication bias, as these tend to be retrospective studies or secondary analyses of trial data collected for other reasons. This potential source of bias should also be noted.

Response

Thank you for these helpful points. We have edited the 'strengths and weaknesses of the study' section of the manuscript to take into consideration your comments. The text below has now been added to the Discussion:

“The review may be limited because a single researcher reviewed the majority of the ORRCA entries, and it is possible that ORRCA may not have included all relevant articles in the first place. This work may also be at risk of publication bias, as it excluded abstracts. However, inclusion of abstracts may have resulted in a lot of missing data as abstracts cannot contain all the data items we were interested in. Another weakness is that the majority of recruitment studies were retrospective analyses of processes and events that occurred during the host RCT'. These data were not necessarily collected with the intention of evaluating RCT recruitment strategies. This may limit the quality of the data and the utility of the recommendations arising from the included studies.

Finally, a risk of bias assessment of the included recruitment studies was not performed, because only one of the recruitment studies was an RCT (i.e. a randomised, controlled evaluation of a recruitment intervention).

VERSION 2 – REVIEW

REVIEWER	Christopher Jones Cooper Medical School of Rowan University, USA I am or have recently been an investigator on unrelated studies sponsored by AstraZeneca, Roche Diagnostics, and Janssen.
REVIEW RETURNED	28-Sep-2017

GENERAL COMMENTS	Thank you for your thoughtful responses to the feedback provided for your initial submission. The authors have substantially improved both the Discussion and Limitations, and the changes to the Abstract and Tables help to clarify these sections. I am encouraged that the ORCCA database was queried through 2014, rather than until 2014. This still means that the search is fairly dated, but it is an improvement. I still believe that the methods would be strengthened by performing a Medline search of manuscripts published since December 2014, but I understand that the nature of this topic makes that difficult. Similarly, performing a duplicate assessment of each potentially eligible study would strengthen your methods, though again I understand that resource limitations might preclude taking this step. I continue to disagree with the decision to exclude abstracts, and instead think inclusion decisions should be made based on the availability of information rather than the publication type. While I understand that abstracts are less likely to contain all of the information you would generally obtain from a full length publication, my opinion is that including them is important for at least three reasons. 1) In some cases the abstract may in fact contain all the information you need. 2) If the published abstract does not provide sufficient information, you could attempt to contact the abstract authors for the missing details. 3) Even when abstracts do not contain all of the information you would like and the authors are unable or unwilling to provide this information, reporting on the number of abstracts which are relevant but unable to be included due to missing information will provide readers with information about the impact of publication bias on your research question. Similarly, I would like to know how many authors you attempted to contact to gain additional information for an eligible study, but were unable to do so and therefore excluded the study in question.
--

VERSION 2 – AUTHOR RESPONSE

We thank you again for further comments on this manuscript and we are pleased that this paper is now recommended for publication. We have responded to the remaining comments below and enclose an edited manuscript with the changes highlighted in yellow.

Reviewer Name: Christopher Jones

1. “Thank you for your thoughtful responses to the feedback provided for your initial submission. The authors have substantially improved both the Discussion and Limitations, and the changes to the Abstract and Tables help to clarify these sections.”

Response

We are grateful for the opportunity to improve the manuscript as recommended.

2. “I am encouraged that the ORCCA database was queried through 2014, rather than until 2014. This still means that the search is fairly dated, but it is an improvement. I still believe that the methods would be strengthened by performing a Medline search of manuscripts published since December 2014, but I understand that the nature of this topic makes that difficult.

Similarly, performing a duplicate assessment of each potentially eligible study would strengthen your methods, though again I understand that resource limitations might preclude taking this step.”

Response

We appreciate that the reviewer understands that further searching in Medline since December 2014 would have been very challenging and that there are no longer resources available to allow for a full duplicate assessment of each potentially eligible study.

3. I continue to disagree with the decision to exclude abstracts, and instead think inclusion decisions should be made based on the availability of information rather than the publication type. While I understand that abstracts are less likely to contain all the information you would generally obtain from a full length publication, my opinion is that including them is important for at least three reasons. 1) In some cases the abstract may in fact contain all the information you need. 2) If the published abstract does not provide sufficient information, you could attempt to contact the abstract authors for the missing details. 3) Even when abstracts do not contain all the information you would like and the authors are unable or unwilling to provide this information, reporting on the number of abstracts which are relevant but unable to be included due to missing information will provide readers with information about the impact of publication bias on your research question.

Response

It has been acknowledged that this is a limitation of the paper, however, other systematic reviews in this area excluded published abstracts (1-14) and this is common practice in many systematic reviews. We chose not to include abstracts because we anticipated that these were unlikely to require the details necessary to make a meaningful contribution to the review. However, we have re-emphasised (in the Discussion) that this decision may have resulted in the omission of potentially useful information. In addition, there is evidence that information from abstracts may change between the abstract publication and full paper –there is therefore a risk that using abstracts will introduce incomplete data sets and less reliable information.(15,16)

The 3rd paragraph of the ‘strengths and weaknesses of this study’ section but has been further amended to now read:

“This work may also be at risk of publication bias, as we chose to exclude abstracts based on the assumption that these were unlikely to include all the data items we were interested in. Excluding abstracts may have resulted in omission of some potentially valuable information. However, there is some evidence to suggest that there is discordance between the content of abstracts and the subsequent full-text publication, and as such, including abstracts may have introduced unreliable data.(64,65)”

4. Similarly, I would like to know how many authors you attempted to contact to gain additional information for an eligible study, but were unable to do so and therefore excluded the study in question.

Response

The number of full-text papers which we were not able to be included due to an inability to contact the study authors was 1. This is shown in the PRISMA diagram for the review. We were able to contact 12 other study authors.

References

1. Boland J, Currow DC, Wilcock A, Tieman J, Hussain JA, Pitsillides C, et al. A systematic review of strategies used to increase recruitment of people with cancer or organ failure into clinical trials: implications for palliative care research. *Journal of pain and symptom management*. 2015;49(4):762-72.e5.
2. Fayter D, McDaid C, Eastwood A. A systematic review highlights threats to validity in studies of barriers to cancer trial participation. *Journal of clinical epidemiology*. 2007;60(10):990-1001.
3. Mc Daid C, Hodges Z, Fayter D, Stirk L, Eastwood A. Increasing participation of cancer patients in randomised controlled trials: a systematic review. *Trials*. 2006;7:16.
4. Ngune I, Jiwa M, Dadich A, Lotriet J, Sriram D. Effective recruitment strategies in primary care research: a systematic review. *Quality in primary care*. 2012;20(2):115-23.
5. Auster J, Janda M. Recruiting older adults to health research studies: A systematic review. *Australasian journal on ageing*. 2009;28(3):149-51.
6. Provencher V, Mortenson WB, Tanguay-Garneau L, Belanger K, Dagenais M. Challenges and strategies pertaining to recruitment and retention of frail elderly in research studies: a systematic review. *Archives of gerontology and geriatrics*. 2014;59(1):18-24.
7. Ibrahim S, Sidani S. Strategies to recruit minority persons: a systematic review. *Journal of immigrant and minority health / Center for Minority Public Health*. 2014;16(5):882-8.
8. Caldwell PH, Hamilton S, Tan A, Craig JC. Strategies for increasing recruitment to randomised controlled trials: systematic review. *PLoS medicine*. 2010;7(11):e1000368.
9. Fletcher B, Gheorghe A, Moore D, Wilson S, Damery S. Improving the recruitment activity of clinicians in randomised controlled trials: a systematic review. *BMJ open*. 2012;2(1):e000496.
10. Huynh L, Johns B, Liu SH, Vedula SS, Li T, Puhan MA. Cost-effectiveness of health research study participant recruitment strategies: a systematic review. *Clinical trials (London, England)*. 2014;11(5):576-83.
11. Mapstone J, Elbourne D, Roberts I. Strategies to improve recruitment to research studies. *The Cochrane database of systematic reviews*. 2007(2):Mr000013.
12. Raftery J, Bryant J, Powell J, Kerr C, Hawker S. Payment to healthcare professionals for patient recruitment to trials: systematic review and qualitative study. *Health technology assessment (Winchester, England)*. 2008;12(10):1-128, iii.
13. Treweek S, Mitchell E, Pitkethly M, Cook J, Kjeldstrom M, Taskila T, et al. Strategies to improve recruitment to randomised controlled trials. *The Cochrane database of systematic reviews*. 2010(1):Mr000013.
14. Watson JM, Torgerson DJ. Increasing recruitment to randomised trials: a review of randomised controlled trials. *BMC medical research methodology*. 2006;6:34.
15. Tam VC, Hotte SJ. Consistency of phase III clinical trial abstracts presented at an annual meeting of the American Society of Clinical Oncology compared with their subsequent full-text publications. *J Clin Oncol*. 2008 May 1;26(13):2205-11.

16. Toma M, McAlister FA, Bialy L, et al. Transition from meeting abstract to full-length journal article for randomized controlled trials. JAMA. 2006;295(11):1281–7.

VERSION 3 – REVIEW

REVIEWER	Christopher Jones Cooper Medical School of Rowan University, USA
REVIEW RETURNED	26-Oct-2017

GENERAL COMMENTS	I appreciate the authors' point of view with respect to including so-called grey literature sources such as conference abstracts in their systematic review, but it seems as if we will continue to disagree over the appropriateness of excluding data from these sources, and leave it to the editors to come to a decision. While I certainly understand that previous authors have chosen to exclude these sources as well, that does not make the decision correct. When systematic reviewers assess for the presence of publication bias, they frequently find it.(1) Not surprisingly, effect estimates often systematically differ between grey literature sources and published full-length manuscripts.(2) For these reasons, many (though admittedly not all) experts recommend including abstracts and other unpublished sources when conducting systematic reviews.(2, 3, 4) My opinion is that in this case the assessment of relevant conference abstracts may help to mitigate or at least allow readers to estimate the impact of publication bias on your dataset. This would involve a relatively modest expenditure of resources (your PRISMA flow chart shows that fewer than 101 studies were excluded because they were abstracts), and without doing the analysis we don't know whether these are relevant and if so whether their inclusion would impact your conclusions. (1) Ziai H, Zhang R, Chan AW, Persaud N. Search for unpublished data by systematic reviewers: an audit. BMJ Open. 2017 Oct 6;7(10):e017737. (2) Hopewell S, McDonald S, Clarke M, Egger M. Grey literature in meta-analyses of randomized trials of health care interventions. Cochrane Database Syst Rev. 2007 Apr 18;(2):MR000010 (3) Balshem H, Stevens A, Ansari M, Norris S, Kansagara D, Shamliyan T, Chou R, Chung M, Moher D, Dickersin K. Finding Grey Literature Evidence and Assessing for Outcome and Analysis Reporting Biases When Comparing Medical Interventions: AHRQ and the Effective Health Care Program. 2013 Nov 18. Methods Guide for Effectiveness and Comparative Effectiveness Reviews [Internet]. Rockville (MD): Agency for Healthcare Research and Quality (US); 2008-. Available from http://www.ncbi.nlm.nih.gov/books/NBK174882/
---

	(4) Institute of Medicine (US) Committee on Standards for Systematic Reviews of Comparative Effectiveness Research; Eden J, Levit L, Berg A, et al., editors. Finding What Works in Health Care: Standards for Systematic Reviews. Washington (DC): National Academies Press (US); 2011. Available from: https://www.ncbi.nlm.nih.gov/books/NBK209518/
--	--

VERSION 3 – AUTHOR RESPONSE

Many thanks for your understanding regarding this submission and for providing us with the opportunity to respond and update the manuscript.

We have made the minor alterations as suggested above.